# Maternal and Fetal PI3K-p110α Deficiency Induces Sex-Specific Changes in Conceptus Growth and Placental Mitochondrial Bioenergetic Reserve in Mice

**DOI:** 10.3390/vetsci9090501

**Published:** 2022-09-13

**Authors:** Daniela Pereira-Carvalho, Esteban Salazar-Petres, Jorge Lopez-Tello, Amanda N. Sferruzzi-Perri

**Affiliations:** Department of Physiology, Development and Neuroscience, Centre for Trophoblast Research, University of Cambridge, Cambridge CB2 3EG, UK

**Keywords:** mitochondria, placenta, fetus, sex, signaling, maternal PI3K-p110α, fetal PI3K-p110α

## Abstract

**Simple Summary:**

The placenta is responsible for materno-fetal resource allocation. Placental insufficiency may disrupt fetal growth trajectory, with negative consequences for maternal and fetal health. By manipulating a growth-controlling protein named PI3K-p110α, this work, performed on mice, shows the relevance of the maternal environment and fetal genotype in determining the conceptus growth and mitochondria bioenergetic reserve in the placental transport zone. Interestingly, the disruption of maternal and/or fetal PI3K-p110α exerted these effects in a sex-specific manner, with male fetuses more affected than females. These data highlight the importance of the sex of the fetus and placental mitochondrial function in the regulation of fetal growth.

**Abstract:**

Fetal growth is reliant on placental formation and function, which, in turn, requires the energy produced by the mitochondria. Prior work has shown that both mother and fetus operate via the phosphoinositol 3-kinase (PI3K)-p110α signalling pathway to modify placental development, function, and fetal growth outcomes. This study in mice used genetic inactivation of PI3K-p110α (α/+) in mothers and fetuses and high resolution respirometry to investigate the influence of maternal and fetal PI3K-p110α deficiency on fetal and placental growth, in relation to placental mitochondrial bioenergetics, for each fetal sex. The effect of PI3K-p110α deficiency on maternal body composition was also determined to understand more about the maternal-driven changes in feto-placental development. These data show that male fetuses were more sensitive than females to fetal PI3K-p110α deficiency, as they had greater reductions in fetal and placental weight, when compared to their WT littermates. Placental weight was also altered in males only of α/+ dams. In addition, α/+ male, but not female, fetuses showed an increase in mitochondrial reserve capacity, when compared to their WT littermates in α/+ dams. Finally, α/+ dams exhibited reduced adipose depot masses, compared to wild-type dams. These findings, thus, demonstrate that maternal nutrient reserves and ability to apportion nutrients to the fetus are reduced in α/+ dams. Moreover, maternal and fetal PI3K-p110α deficiency impacts conceptus growth and placental mitochondrial bioenergetic function, in a manner dependent on fetal sex.

## 1. Introduction

In mammals, adequate placental function is required to support fetal growth and development. Evidence from human and experimental animal models have shown that aberrant placental transport function is associated with fetal growth restriction (FGR), a condition that currently affects approximately 5–10% of pregnancies [1,2,3]. Moreover, FGR is associated with an increased risk of perinatal mobility and mortality [4] and poor metabolic health in adulthood [5,6]. Work performed in rodent models exposed to adverse maternal conditions (e.g., maternal undernutrition, hypoxia, or genetic manipulations) has shown that, even though placental development may be compromised, the placenta may maintain, or even increase, nutrient supply to the fetus, thus optimising fetal growth in that prevailing environment [7,8,9,10,11,12,13]. Moreover, the ability of the placenta to functionally adapt in response to an adverse gestational environment has been reported to vary with fetal sex [14,15]. In particular, there are changes in the expression of nutrient transporters, hormone metabolic genes, and growth signalling pathways in the placenta of rodents exposed to maternal hypoxia, endocrine modulation, and suboptimal diets [15,16,17]. However, further work is required to delineate the role of the mother and fetus in mediating placental adaptations and uncover the cellular pathways by which these may occur in female and male fetuses [18].

Several signalling pathways have been described to be involved in the control of placental growth and function [5,13,18,19,20,21]. One key signalling pathway is the phosphoinositol 3-kinase (PI3K) pathway, which is highly conserved in mammals and fine-tuned growth, in accordance with nutrient availability [22,23]. Of key importance is the ubiquitously and most dominantly expressed Class IA PI3K isoform p110α, which signals downstream of insulin. In mice, changes in Class IA PI3K signalling have been reported in maternal and placental tissues, when the ability of the mother or the placenta to provide substrates to the fetus is compromised [8,9,24,25]. Homozygous deficiency of the gene encoding p110α (*Pik3ca*) results in embryonic death, whilst heterozygous deficiency leads to placental dysmorphogenesis, adaptive up-regulation of nutrient transport, and FGR in mice [13,18,20,26]. Mice with a heterozygous deficiency in p110α remain smaller postnatally and exhibit changes in glucose, insulin, and lipid handling prior to and during pregnancy, compared to wild-type (WT) females [18,20]. There are also alterations in placental development and nutrient transport in heterozygous p110α deficient mothers [18]. Together, these data highlight the significance of the mother and fetus operating via PI3K-p110α to modify placental morphogenesis, function, and fetal outcomes.

Whole body metabolism and energy balance has been reported to be affected in genetic models with disrupted PI3K-p110α signalling [27]. In metabolically-active tissues, signalling via p110α plays a role in regulating the ability of the mitochondria to produce energy via oxidative phosphorylation (OXPHOS) [22,28,29]. In addition, there are emerging data showing that placental mitochondrial bioenergetic capacity varies, depending on the sex of the fetus [15,19,21,29]. Therefore, the aim of this study was to determine the influence of maternal and fetal p110α deficiency on fetal and placental growth, in relation to placental mitochondrial bioenergetics for each fetal sex. The effect of p110α deficiency on maternal body composition was also determined, in order to understand more about the maternal-driven changes in conceptus physiology.

## 2. Materials and Methods

### 2.1. Animals and Experimental Design

This study was performed in accordance with the UK Home Office Animals (Scientific Procedures) Act 1986 and University of Cambridge ethics committee. Mice were housed in the University of Cambridge Animal Facility under a 12:12 h light-dark cycle with free access to chow food (Rodent No. 3 breeding chow; Special Diet Services, Witham) and water. Non-pregnant WT and heterozygous PI3K-p110α deficient (α/+) females were reciprocally mated with WT and α/+ male mice to generate pregnancies with both WT and α/+ conceptuses. All mice were on a C57BL/6 background and WT, and α/+ females were nulliparous and 4 months old at the time of mating. The day a copulatory plug was detected was designated gestational day 1 (GD1). The generation of α/+ mice has been previously described [26].

On gestational day 18 (GD18), between 08:00 h and 09:00 h, dams were killed by cervical dislocation. The gravid uterus was removed, and each fetus and its corresponding placenta were weighed. Maternal organs were also dissected and weighed. Fetal tails were taken for sex and genotype determination by PCR. Fetal sex was determined by detection of the *Sry* gene using the Taq Ready PCR system (Sigma) and specific primer sequences (F: 5′-GTGGGTTCCTGTCCCACTGC-3′, R: 5′-GGCCATGTCAAGCGCCCCAT-3′ and F: 5′-TGGTTGGCATTTTATCCCTAGAAC-3′, R: 5′-GCAACATGGCAACTGGAAACA-3′ as a PCR autosomal gene control) and α/+ mice identified by genotyping using primers 5′-TTCAAGCACTGTTTCAGCT-3′ and 5′-TTATGTTCTTGCTCAAGTCCTA-3′, as previously described [20,22]. After weighing the placenta, the labyrinth zone (Lz; transport region) was mechanically separated from the endocrine junctional zone, weighed, and immediately placed in cryopreservation media prior to snap freezing (0.21 M mannitol, 0.07 M sucrose, 30% DMSO, pH 7.5) and storage at −80 °C for subsequent mitochondrial respiratory analyses 1 month later.

### 2.2. Placental Lz Mitochondrial Respirometry

High resolution respirometry (Oxygraph 2k respirometer; Oroboros Instruments, Innsbruck, Austria) was used to assess the placental capacity for mitochondrial OXPHOS, namely substrate use and electron transfer system (ETS) function, as previously described [21]. Briefly, cryopreserved Lz samples were gently thawed in sucrose solution (pH 7.5) and permeabilized in biopsy preservation (BIOPS) solution (pH 7.1, containing 10 mM Ca-EGTA buffer, 0.1 µM free Ca^2+^, 1 mM free Mg^2+^, 20 mM imidazole, 20 mM taurine, 50 mM K-MES, 0.5 mM DTT, 6.56 mM MgCl_2_, and 5.77 mM ATP and 15 mM phosphocreatine) containing saponin (5 mg/mL, Sigma-Aldrich, UK) for 20 min, followed by 3 washes of 5 min in respiratory medium (MiR05; pH 7.1 solution containing 20 mM HEPES, 0.46 mM EGTA, 2.1 mM free Mg^2+^, 90 mM K^+^, 10 mM Pi, 20 mM taurine, 110 mM sucrose, 60 mM lactobionate, and 1 g/L BSA). Samples were blotted on filter paper to determine wet tissue mass, and then 15–20 mg of the Lz sample was placed into a pre-calibrated chamber of an Oxygraph-2k respirometer with respiratory medium MiR05. Samples were analyzed at 37 °C, and oxygen concentration was kept between 250 and 300 μM. DatLab software (V7, Oroboros Instruments) was used for real-time acquisition and analysis of oxygen consumption. A sequential substrate, inhibitor, and uncoupler titration protocol was then performed, as described previously [21]. Addition of cytochrome C (10 µM) at the end of the protocol provided information on the integrity of mitochondrial membranes. Samples were excluded if respiration increased by >30% (a total of 7 placentas were excluded). Respiratory rates were expressed as oxygen consumption per mg of wet mass of placental Lz tissue. Details of the parameters measured, including calculations performed, are shown in Table 1.

## 3. Statistical Analysis

Statistical analyses were performed using GraphPad Prism version 9 (GraphPad, CA, USA). Maternal body composition and litter size composition (sex and genotype) were analysed using Student *t*-test or Mann–Whitney test, based on a prior analysis of the normality and homogeneity of all variables assessed using the Shapiro-Wilk test. The effect and interaction of maternal and fetal α/+ genotype was evaluated by two-way ANOVA, followed by Tukey *post hoc* tests, for each fetal sex separately. Litter composition (sex and genotype) was analyzed by student *t*-test. Outliers were detected by the Prisms Grubbs’ test. Data are shown as mean ± SEM and individual points when possible. Data were considered statistically significant at values of *p* < 0.05.

## 4. Results

### 4.1. PI3K-p110α Deficiency Affects Maternal Body Composition

Female α/+ mice were lighter at the start of the pregnancy and showed a tendency for lighter hysterectomised weight at the end of the pregnancy (Table 2). Pregnant α/+ mice also had lighter gonadal and mesenteric fat pads, whilst the weight of other fat pads and other organs, namely the kidneys and spleen, were not different from WT pregnant females (Table 2). Thus, p110α deficiency has specific impacts on maternal adipose tissue stores in late pregnancy.

### 4.2. Fetal and Maternal PI3K-p110α Deficiency Induces Sex-Specific Changes in Feto-Placental Growth

There was no difference in litter size, genotype, or sex frequency at GD18 between the WTxα/+ and α/+xWT parental crosses (Table 3). There was an overall effect of fetal α/+ to reduce the weight of female fetuses by 11%, regardless of maternal genotype (Figure 1A). The effect of fetal α/+ was stronger in males, as α/+ males were significantly smaller than WT littermates, irrespective of maternal genotype (−23%). Whereas, both female and male WT and α/+ fetuses developed in a α/+ mother were significantly smaller (−17% females and −15% males), compared to WT and α/+ fetuses carried by WT dams (Figure 1A).

In females, there was not an effect of fetal or maternal α/+ on placental weight (Figure 1B). However, placentas from α/+ males were lighter than WTs, an effect that was significant in α/+ mothers (−22%; *p* < 0.001; Figure 1B). Moreover, male placentas in α/+ mothers were overall heavier, compared to those from WT mothers, an effect that was significant by pairwise comparison for WT fetuses (+18%, Figure 1B). There was an overall effect of fetal α/+ to reduce Lz weight for both female (−12%) and male fetuses (−24%; Figure 1C). In the case of males, the effect of fetal α/+ to decrease Lz weight was significant by pairwise comparisons in α/+ dams (−32%; *p* < 0.001), and this was reflected by an interaction between fetal and maternal α/+ genotype. In female fetuses only, there was an overall effect of maternal α/+ to reduce Lz weight by 11% (Figure 1C). There was no effect of fetal or maternal α/+ genotype on placental Lz efficiency (defined as the ratio between fetal weight and Lz weight) (Figure 1D). However, placental Lz efficiency was lower for males in α/+ dams, with pairwise comparisons identifying a significant effect on WTs (−28%; *p* < 0.001). Together, these data indicate that the effect of fetal and maternal p110α deficiency on fetal-placental growth is modified by fetal sex, with more dramatic effects observed in male fetuses.

### 4.3. Fetal and Maternal PI3K-p110α Deficiency Has Only a Minor Effect on Lz Mitochondrial Respiratory Capacity and Does So in a Sex-Specific Manner

Accounting for the changes in feto-placental growth, the effect of maternal and fetal α/+ to induce alterations in the placental Lz mitochondria function of female and male fetuses was then evaluated. Surprisingly, no major changes in Lz oxygen consumption, due to fetal or maternal p110α deficiency in either sex, were detected, with the exception that mitochondrial reserve capacity (calculated as Total ETS-CI + CII_OXPHOS_) was significantly greater for male α/+ fetuses, compared to their respective WT littermates in α/+ dams (+235%, *p* = 0.0388; Figure 2A). Moreover, there was a tendency for an interaction between the fetal and maternal α/+ genotype for maximum mitochondrial ETS capacity (Total ETS) in females (P_Int_ = 0.0642) and CI + CII_P_ OXPHOS in males (P_Int_ = 0.0628) (Figure 2B,C, respectively). There was no effect of fetal or maternal p110α deficiency on any other respirometry parameter measured (CI LEAK, CII OXPHOS, FAO, CI LEAK or OXPHOS flux control ratios, or CIV activity) (Figure 2). These results indicate that, despite changes in feto-placental growth, fetal and maternal p110α deficiency has only modest sex-related effects on Lz mitochondrial respiration.

## 5. Discussion

Using genetic manipulations, this study demonstrates that a deficiency in PI3K-p110α signalling in the mother and/or fetus modified, in a sex-specific manner, the conceptus mass and mitochondrial bioenergetic capacity in the mouse placental Lz. For instance, male fetuses were more sensitive than females to fetal PI3K-p110α deficiency, as they presented less fetal, placental, and Lz weight, when compared to their WT littermates. Placental weight was also altered in males only of α/+ dams. In addition, α/+ male fetuses showed an increase in Lz mitochondrial reserve capacity, when compared to their WT male littermates in α/+ dams. Sexual dimorphism, in response to genetic and/or gestational perturbations in placental function, has been extensively reported in the recent years [30,31,32,33]. This study provides valuable information on how signalling via PI3K-p110α is important for mediating both intrinsic and extrinsic impacts on the placenta, as well as its relevance for female and male fetal growth.

Previous data reported that PI3K-p110α deficiency in the fetus negatively affects fetal growth and placental development [13,18]. This study shows that this effect is most pronounced for the males, who showed a ~15–25% reduction in fetal weight, as well as lighter placentas with a smaller Lz. Whether the Lz morphological defects previously reported for α/+ fetuses are also only seen for males [13,18] requires further study. Reductions in placental Lz size was greatest in male α/+ fetuses from α/+ dams. Despite this, only male α/+ fetuses from α/+ dams presented an increased reserve capacity, in line with the observed statistical tendency for the fetal and maternal p110α genotype to interact and determine the (CI + CII)_OXPHOS_-related oxygen flux. The mitochondrial reserve capacity is a parameter used to describe the amount of extra ATP that can be produced by OXPHOS, in case of a sudden increase in energy demand [34]. This increase in reserve capacity seems to be related to an apparent tendency for reduced (CI + CII)OXPHOS-related oxygen flux in Lz from α/+ male fetuses. Impaired CI- and CII-related respiration is frequently described in diseases such as obesity and diabetes [35]. Interestingly, male guinea pig placentas subjected to chronic hypoxia show inhibited mitochondrial complexes I and IV, and this effect was not seen in females [36]. There are also moderate effects of other gestational manipulations, e.g., maternal obesity on placental mitochondrial function that are partly influenced by fetal sex [15,37]. Together these data suggest that the metabolic adaptation of placental Lz mitochondria in male α/+ fetuses would enable them to better draw on their reserve capacity and meet increasing bioenergetic needs, which may be particularly important, given their growth-compromised placental Lz. There was also a tendency for an interaction between the fetal and maternal p110α genotype in dictating the total ETS of the placental Lz for female fetuses only. Together, these data reinforce the idea of a sex-specific response of placental mitochondrial respiratory capacity, which is consistent with recent work published on mice [15,20] or trophoblast [38] isolated from the human placenta, where efforts have been made to compare males and females. Whether these findings have relevance for understanding the different age-related metabolic outcomes of α/+ male and female mice postnatally, which have previously [39] been reported, is unclear.

Previous data has also identified that placental development is influenced by whether the mother herself carries the p110α mutation (α/+) [18]. Placental and Lz weights were greater specifically for males of α/+ dams, an effect most evident for the WT fetuses. Thus, α/+ placentas are unable to adapt. Fetal weight was, however, less for both females and males in α/+ dams, regardless of whether they were WT or α/+. Moreover, prior work has reported that pregnant α/+ dams show reduced liver and pancreas weights, elevated circulating leptin and insulin concentrations, and increased insulin sensitivity, when compared to WT dams that had been mated to either WT males or α/+ males [18]. Here, this study finds that maternal PI3K-p110α deficiency affects the size of specific maternal adipose tissues in late pregnancy, with reduced sizes of the gonadal and mesenteric fat depots in α/+ dams mated to WT males. These data indicate that both maternal nutrient reserves and the ability to apportion nutrients to the fetus are reduced in α/+ dams. Moreover, the current study shows that the nature of the changes observed in conceptus outcome depends, in part, on whether the fetus was male or female. It may be that male fetuses, which have a greater intrinsic drive for growth than females [21], are more negatively affected by this reduced ability of the α/+ dam to support the pregnancy.

## 6. Conclusions

In summary, this work, using genetic manipulations of the growth and metabolic regulator PI3K-p110α, has established that changes in the fetal genetic ability to grow, as well as the genetically-determined maternal environment, affect feto-placental development and placental respiratory function in a sex-specific manner. Moreover, the effects detected in mitochondria respiratory function seem to depend on an interaction between the mother and/or fetus carrying the mutation for each sex. These findings may have importance for understanding the fetal outcomes in pregnancies reporting alterations in the PI3K-AKT signalling pathway in the placenta [40,41], and/or the mother [42,43,44], as well as for growth dimorphism between male and female fetuses more generally. They also highlight the relevance of mitochondria in mediating genotypic and environmental influences on the placental support of male and female fetal growth and suggest that agents targeting placenta mitochondria may be beneficial in optimising fetal outcomes [45,46].

## Figures and Tables

**Figure 1 vetsci-09-00501-f001:**
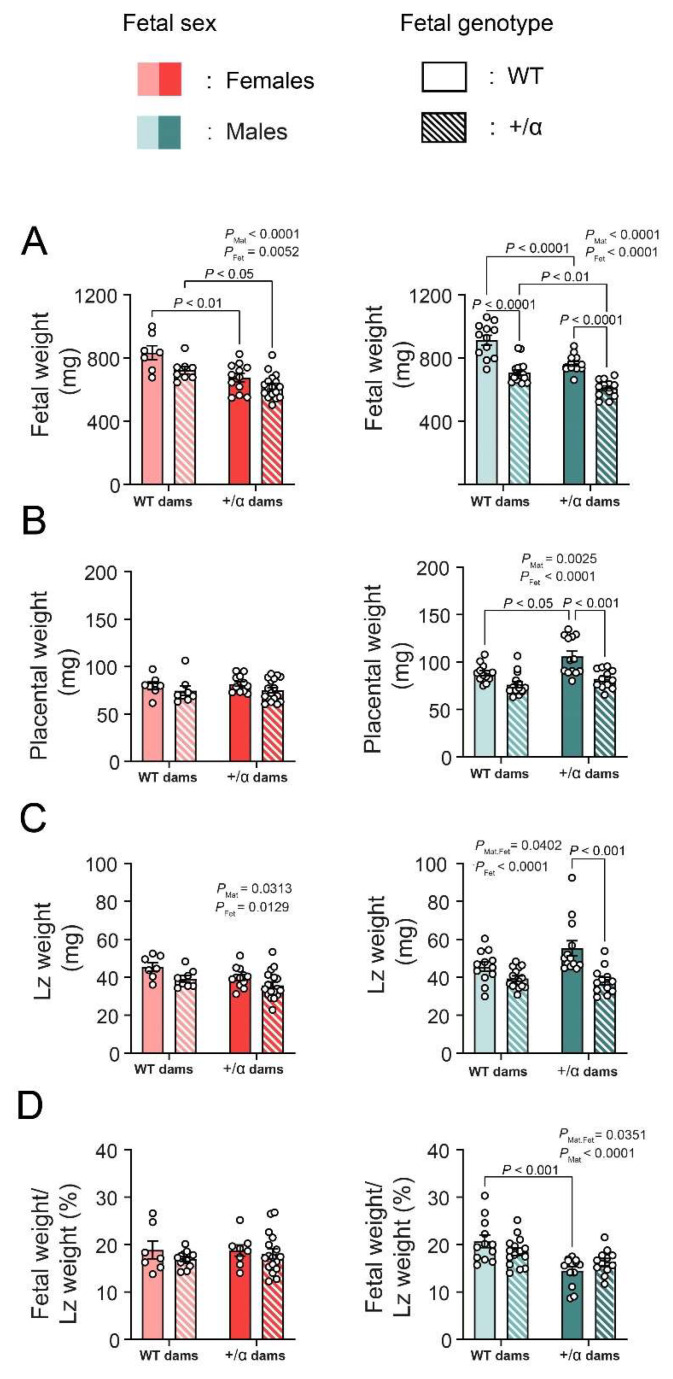
**Fetal-placental growth in response to fetal or maternal p110α deficiency in the two fetal sexes.** Fetal weight (**A**), placental weight (**B**), Lz weight (**C**), and fetal weight/Lz weight (**D**) in females and males on day 18 of pregnancy. Data are displayed as individual data points with mean ± SEM. Data were analyzed by two-way ANOVA with Tukey *post hoc* pairwise comparisons. Lz: labyrinth zone.

**Figure 2 vetsci-09-00501-f002:**
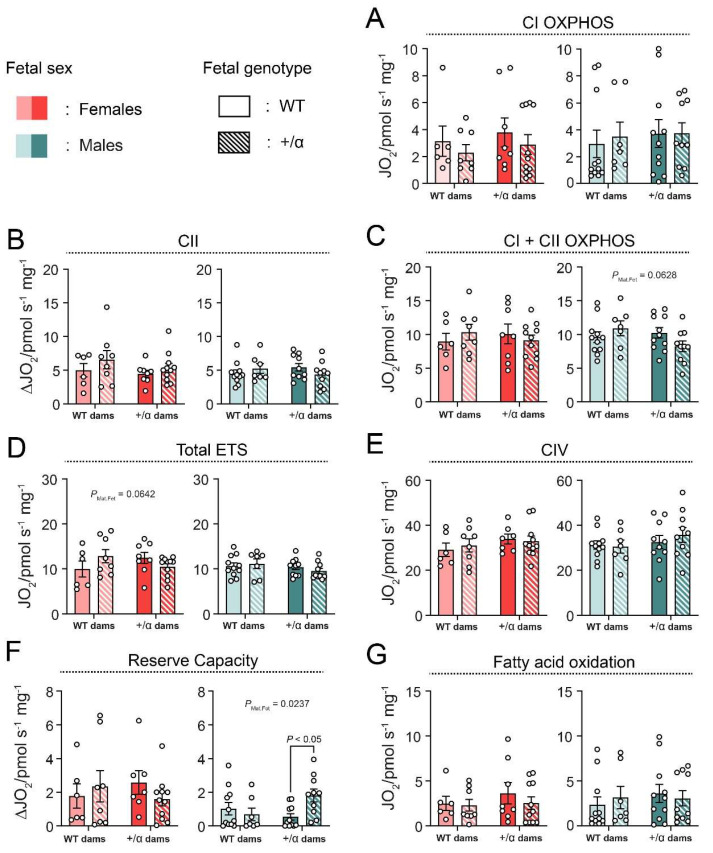
Placental labyrinth mitochondrial respiration in response to fetal or maternal p110α deficiency in the two fetal sexes. Mitochondrial complexes I and II substrate-driven respiration (**A**–**C**), maximum electron transfer system capacity (total ETS) (**D**), mitochondrial complex IV substrate-driven respiration (**E**), reserve capacity (**F**), and total fatty acid oxidation (**G**). Data are from all fetuses generated by female x male parental crosses, i.e., WTxα/+ and α/+xWT, and displayed as individual data points with mean ± S.E.M. Data were analyzed by two-way ANOVA with Tukey *post hoc* pairwise comparisons.

**Table 1 vetsci-09-00501-t001:** List of the parameters measured during the substrate, inhibitor, and uncoupler titration assay for mitochondrial respirometry using an Oroboros oxygraph.

Parameter Measured	Function	Reagents	Required Calculations
Complex I _OXPHOS_	O_2_ consumption linked to ATP synthesis via Complex I	Pyruvate (20 mM) and glutamate (10 mM)	Raw O_2_ consumption after glutamate addition
Complex II	O_2_ consumption linked to ATP synthesis via Complex II	Malonate (1 µM)	Difference between O_2_ consumption before and after adding malonate
Complex I + II _OXPHOS_	Complex I and II dependent oxidative phosphorylation	Succinate (10 mM)	Raw O_2_ consumption after succinate addition
Total ETS	Maximal uncoupled ETS-respiratory capacity	Trifluoromethoxy carbonyl-cyanide phenylhydrazone (FCCP, 3 doses of 0.5 mM)	Raw O_2_ consumption after 2 doses of FCCP addition
Complex IV	Complex IV activity	Sodium ascorbate (2 mM), N, N, N’, N’-tetramethyl-p-phenylenediamine (TMPD, 0.5 mM), and sodium azide (200 mM).	Correction for chemical background oxygen consumption in the presence of sodium azide after sodium ascorbate and TMPD addition
Reserve capacity	Mitochondrial capacity to produce extra ATP by OXPHOS	Succinate (10 mM) and FCCP (3 doses of 0.5 mM)	The difference between total ETS and CI + IIP values: Reserve = Total ETS—(CI + II)P
FAO	Fatty acid oxidation	ADP (5 mM)	Raw O_2_ consumption after ADP addition

**Table 2 vetsci-09-00501-t002:** **The effect of PI3K-p110α deficiency on maternal body weight.** Data were obtained on GD18, except for starting weight, which corresponded to GD1. Ratios were obtained by dividing the absolute weight of the organ by the hysterectomised weight of the dam. Statistical analysis performed by Student *t*-test or Mann–Whitney test (spleen weight and spleen ratio), based on the normality of the variable.

	WT Female × α/+ Male (*n* = 5)	α/+ Female × WT Male (*n* = 7)	*p* Value
Starting weight (g)	22.7 ± 0.40	20.2 ± 0.54	0.006
Hysterectomy weight (g)	24 ± 1.26	21.4 ± 0.67	0.08
Absolute weights			
Gonadal fat (mg)	505 ± 31	318 ± 43.5	0.009
Retroperitoneal fat (mg)	78.3 ± 9.29	58.1 ± 9.50	0.17
Renal fat (mg)	111 ± 18.60	97.4 ± 19.40	0.64
Mesenteric fat (mg)	267 ± 28.7	136 ± 15.8	0.001
Subcutaneous inguinal fat (mg)	444 ± 26.7	380 ± 28.7	0.14
Subcutaneous dorsal fat (mg)	299 ± 36.5	236 ± 31.2	0.21
Kidneys (mg)	279 ± 7.97	246 ± 14.5	0.11
Spleen (mg)	70.8 ± 2.54	90.7 ± 15.1	0.30
Ratios			
Gonadal fat (%)	2.12 ± 0.15	1.47 ± 0.17	0.025
Retroperitoneal fat (%)	0.32 ± 0.03	0.26 ± 0.03	0.28
Renal fat (%)	0.46 ± 0.08	0.44 ± 0.07	0.85
Mesenteric fat (%)	1.10 ± 0.07	0.62 ± 0.05	0.0005
Subcutaneous inguinal fat (%)	1.87 ± 0.12	1.76 ± 0.09	0.50
Subcutaneous dorsal fat (%)	1.25 ± 0.13	1.09 ± 0.11	0.38
Kidneys (%)	1.17 ± 0.04	1.15 ± 0.03	0.68
Spleen (%)	0.30 ± 0.02	0.42 ± 0.07	0.20

**Table 3 vetsci-09-00501-t003:** **Litter size and composition.** Crosses are shown as female parent x male parent. Data are from 5–7 dams per group on Gd18 and presented as mean ± SEM. Data were analyzed using the unpaired Student’s *t*-test.

	WT Female × α/+ Male (*n* = 5)	α/+ Female × WT Male (*n* = 7)	*p* Value
**Litter size**	7.20 ± 0.99	7.27 ± 0.57	0.85
**% Females**	41.67 ± 0.35	55.77 ± 0.34	0.28
**% Males**	58.33 ± 0.36	44.23 ± 0.36	0.41
**% α/+ Females**	18.69 ± 6.58	28.57 ± 8.32	0.42
**% WT Females**	25.63 ± 9.96	24.29 ± 4.88	0.91
**% α/+ Males**	22.74 ± 7.14	17.43 ± 4.92	0.59
**% WT Males**	32.94 ± 2.93	29.71 ± 10.43	0.79

## Data Availability

All relevant data are within the paper and available upon reasonable request.

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
