# Peer review of "Maternal and Fetal PI3K-p110α Deficiency Induces Sex-Specific Changes in Conceptus Growth and Placental Mitochondrial Bioenergetic Reserve in Mice"

_vetsci, 2022, doi:10.3390/vetsci9090501_

Round 1

Reviewer 1 Report

I think that the subject of the work is of interest and that the topic of the manuscript is appropriate for the Journal. The information is of significant interest to the Journal's readers. 

The title accurately reflects the major findings of the work. The abstract adequately summarize methodology, results, and significance of the study. However, Authors should indicate statistical analysis applied and report the p value when significant differences have found. Please avoid the use of personal form (our, we etc.)

Keywords represent the article adequately. However, I suggest to change “P13K” with “maternal PI3K-p110α” and “fetal PI3K-p110α”

I suggest to avoid the use of personal form throughout the text.

The introduction section is well written and it falls within the topic of the study. Authors cited appropriately references on the topic of the study.

The section of Material and Methods is clear for the reader and it meticulously describes the methods applied in the study.  However, Authors should specify the full term of the acronyms which appear for the first time. Regarding sampling, Authors should specify the time between sampling and laboratory analysis. Authors should indicate whether the sampling was performed at the same hour of the day in order to exclude any influence of circadian rhythm.

Regarding statistical analysis Authors wrote that a normality test on data in order to assess their normal distribution was applied. Please add the results of the analysis with related P value.

Results section as well as Discussion section is well written. The findings obtained in the study were well discussed and justified with appropriate references.

I suggest to add a separated conclusion section  to better summarize the results and to emphasize the significance of the study.

The tables and figures are generally good and well represent the results of the study.

Authors should check and standardize the references in the list according to journal guidelines.

Author Response

I think that the subject of the work is of interest and that the topic of the manuscript is appropriate for the Journal. The information is of significant interest to the Journal's readers. 

R: we appreciate the reviewer's comment and the time spent reviewing our manuscript.

 The title accurately reflects the major findings of the work. The abstract adequately summarize methodology, results, and significance of the study. However, Authors should indicate statistical analysis applied and report the p value when significant differences have found. Please avoid the use of personal form (our, we etc.)

R: p-values and statistical analysis performed were now indicated in the text and figure legends. Moreover, we have modified the figures and the tables so the reader can see the exact p value.

 Keywords represent the article adequately. However, I suggest to change “P13K” with “maternal PI3K-p110α” and “fetal PI3K-p110α”

R: keywords were added as suggested by the reviewer.

 I suggest to avoid the use of personal form throughout the text. The introduction section is well written and it falls within the topic of the study. Authors cited appropriately references on the topic of the study.

R: As suggested, we have removed the personal forms throughout the text.

 The section of Material and Methods is clear for the reader and it meticulously describes the methods applied in the study.  However, Authors should specify the full term of the acronyms which appear for the first time.

R: All acronyms are now defined when they first appear in the revised paper.

 Regarding sampling, Authors should specify the time between sampling and laboratory analysis. Authors should indicate whether the sampling was performed at the same hour of the day in order to exclude any influence of circadian rhythm.

R: Time between sampling and lab analysis was consistent across the study. This information can now be found on lines 99 and 112.

 Regarding statistical analysis Authors wrote that a normality test on data in order to assess their normal distribution was applied. Please add the results of the analysis with related P value.

R: We appreciate the comment of the reviewer. We have clarified this aspect in the section of statistical analysis. We run the normality test for the maternal data and only two parameters did not follow the normality (spleen weight and spleen ratio). In the legend for Table 2, we have added additional information. We have attached here the P values so the reviewer can see them. However, the P values for the Shapiro-Wilk test are not normally shown in the papers. We believe that reporting the parameters that were run by t-test versus those done with Mann-Whitney test is clear enough (as stated in the new legend). However, if the reviewer consider that these P values need to be shown, we will include then in the second revisions as supplementary information. We hope the reviewer sympathise with our reply. 

Parameter

P value

Litter size

0.6833

Starting weight (g)

0.7301

Hysterectomy weight (g)

0.1822

Absolute weights

Gonadal fat (mg)

0.5902

Retroperitoneal fat (mg)

0.9522

Renal fat (mg)

0.1536

Mesenteric fat (mg)

0.4839

Subcutaneous inguinal fat (mg)

0.8213

Subcutaneous dorsal fat (mg)

0.7479

Kidneys (mg)

0.0773

Spleen (mg)

0.0153

Ratios

Gonadal fat (%)

0.7041

Retroperitoneal fat (%)

0.8854

Renal fat (%)

0.3514

Mesenteric fat (%)

0.8813

Subcutaneous inguinal fat (%)

0.2279

Subcutaneous dorsal fat (%)

0.2439

Kidneys (%)

0.1909

Spleen (%)

0.0057

Results section as well as Discussion section is well written. The findings obtained in the study were well discussed and justified with appropriate references. I suggest to add a separated conclusion section  to better summarize the results and to emphasize the significance of the study.

R: We have now separated the final summary paragraph from the main discussion, by placing it beneath the subsection ‘Conclusions’. We have also included a few more details in the conclusion, but kept this to a minimum to avoid repetition of the results already discussed and to comply with the short article format for the journal. Please see Lines 263-275 for the next inserted text.

The tables and figures are generally good and well represent the results of the study.

Authors should check and standardize the references in the list according to journal guidelines.

R: References have been double checked and standardized.

Reviewer 2 Report

In this manuscript entitled “Maternal and fetal PI3K-p110a deficiency induce sex-specific changes in conceptus growth and placental mitochondrial bioenergetic reserve”, the authors examine the fetal and placental growth from p110a heterozygous mice. The authors also investigate the effects of placental Lz mitochondrial function.

Overall, the observations of the study are interesting; however, the present manuscript represents a more limited advance than the previous paper published by the same group. The problem with this manuscript is that the "Discussion" section is based on the group's previous papers. The authors should address the multifaceted interpretation of the negative results by citing many internal and external references.

Author Response

In this manuscript entitled “Maternal and fetal PI3K-p110a deficiency induce sex-specific changes in conceptus growth and placental mitochondrial bioenergetic reserve”, the authors examine the fetal and placental growth from p110a heterozygous mice. The authors also investigate the effects of placental Lz mitochondrial function.

Overall, the observations of the study are interesting; however, the present manuscript represents a more limited advance than the previous paper published by the same group.

Comments:

  1. Discussion: It is not clearly stated why the effect on mitochondrial respiration was minimal. The authors should address the multifaceted interpretation of the modest effects on mitochondrial respiration by citing many internal and external references.

R: We would like to thank you for the time spent reviewing our manuscript. We agree with the comment and in this last version, we have adapted the discussion on mitochondrial respiration as well as included additional references. Please see Lines 227-232 and Lines 264-266.

Reviewer 3 Report

The manuscript presents novel and interesting information. However, some corrections should be made:

English language should be checked. There are some spelling and minor errors that must be corrected. For example, the title should say induces, not induce.

The fluency of the writing is challenging to follow, and it does not include some essential details. For example, in the abstract, the species used for the study is not mentioned but should be stated. Similar comments could be valid for the aim where the species evaluated is not there; the reason why that may indicate that the aim means a general statement for all the species.

As a comment, I prefer to use more impersonal writing in a scientific paper avoiding terms such as our or we.

Author Response

 English language should be checked. There are some spelling and minor errors that must be corrected. For example, the title should say induces, not induce.

 R: Amended and checked throughout.

The fluency of the writing is challenging to follow, and it does not include some essential details. For example, in the abstract, the species used for the study is not mentioned but should be stated. Similar comments could be valid for the aim where the species evaluated is not there; the reason why that may indicate that the aim means a general statement for all the species.

R: Apologies for this omission. Mice have now been added in lines 2 and 23.

As a comment, I prefer to use more impersonal writing in a scientific paper avoiding terms such as our or we.

R: As suggested, we have amended all sentences so that the writing is impersonal.